# Glutamine metabolism modulates chondrocyte inflammatory response

**Manoj Arra[1], Gaurav Swarnkar[1], Naga Suresh Adapala[1], Syeda Kanwal Naqvi[1], Lei Cai[1], Muhammad Farooq Rai[1], Srikanth Singamaneni[2], Gabriel Mbalaviele[3], Robert Brophy[1], Yousef Abu-Amer[1,4]\***

[1]Department of Orthopedic Surgery, Washington University School of Medicine, St Louis, United States; [2]Department of Mechanical Engineering and Material Sciences, Washington University School of Medicine, St Louis, United States; [3]Bone and Mineral Division, Department of Medicine, Washington University School of Medicine, St Louis, United States; [4]Shriners Hospital for Children, Saint Louis, United States

**Abstract** Osteoarthritis is the most common joint disease in the world with significant societal consequences but lacks effective disease-modifying interventions. The pathophysiology consists of a prominent inflammatory component that can be targeted to prevent cartilage degradation and structural defects. Intracellular metabolism has emerged as a culprit of the inflammatory response in chondrocytes, with both processes co-regulating each other. The role of glutamine metabolism in chondrocytes, especially in the context of inflammation, lacks a thorough understanding and is the focus of this work. We display that mouse chondrocytes utilize glutamine for energy production and anabolic processes. Furthermore, we show that glutamine deprivation itself causes metabolic reprogramming and decreases the inflammatory response of chondrocytes through inhibition of NF-κB activity. Finally, we display that glutamine deprivation promotes autophagy and that ammonia is an inhibitor of autophagy. Overall, we identify a relationship between glutamine metabolism and inflammatory signaling and display the need for increased study of chondrocyte metabolic systems.

*For correspondence:
abuamery@wustl.edu

Competing interest: The authors declare that no competing interests exist.

## Editor's evaluation

This manuscript focuses on identifying how metabolism can influence the response of cartilage cells to inflammation. This has a relevance to the painful disease known as osteoarthritis. Modulation of call metabolism in the right direction can serve to protect joint cartilage from the negative effects of inflammation with causes onset and disease progression.

## Introduction

Joint disease afflicts millions of individuals around the world, though these conditions are greatly understudied. While many different diseases can affect the joints, osteoarthritis (OA) is the most common, affecting over 200 million individuals (*Allen et al., 2022*). Significant advancements have been made in the treatment of classical inflammatory joint diseases, such as rheumatoid arthritis or psoriatic arthritis, due to identification of therapeutic targets and biomarkers (*Palfreeman et al., 2013*; *Shams et al., 2021*). Disease modifying compounds, especially with the advent of biologics, have revolutionized the management of these patients and allowed improved joint outcomes. However, OA still lacks disease modifying interventions, leading to significant medical and financial burden in the United States and around the globe. (*Bedenbaugh et al., 2021*; *Zhao et al., 2019*). OA can affect many different joints and presents in a variety of manners, likely contributing to the lack of clear disease pathophysiology or therapies, though inflammatory and biomechanical factors

play a role, amongst other factors (*Deveza and Loeser, 2018*; *Mobasheri and Batt, 2016*). OA is characterized by joint degradation, with articular cartilage damage and joint space narrowing. Clinically, patients have increased pain and loss of joint mobility that can progress to significantly impair functionality. Interventions include use of NSAIDs and intra-articular steroid injections for pain relief as well as joint replacement surgeries. However, there are currently no medications for preventing or reversing joint damage caused by OA, and several clinical trials have failed (*Hermann et al., 2018*). Furthermore, the disease is indolent and slowly progressing, often presenting with its classical symptoms long after joint degradation has begun and usually long after inciting factors, such as joint injury, have taken place (*Blasioli and Kaplan, 2014*; *Roos and Arden, 2016*). Clearly, there is a need for novel biomarkers, therapeutic targets, and overall understanding of disease pathophysiology.

Various groups have shown that inflammation is a driver of OA, even though it may not present like traditional inflammatory diseases such as rheumatoid arthritis (*Arra et al., 2022*; *Arra et al., 2020*; *Goldring and Otero, 2011*). Joint inflammation causes cartilage-resident chondrocytes, as well as other joint infiltrating cells, to generate catabolic enzymes to promote an overall joint degradative state (*Blasioli and Kaplan, 2014*). Inflammatory stimuli activate signaling pathways such as the NF-κB pathway, which are important drivers of OA disease but have not been successfully targeted in OA (*Arra et al., 2022*; *Arra et al., 2020*; *Catheline et al., 2021*; *Choi et al., 2019*). These stimuli can range from cytokines to mechanical stress to inorganic particulate matter, making it difficult to target specific inflammatory mediators (*Liu-Bryan and Terkeltaub, 2015*; *van den Bosch, 2019*). Due to this, it is necessary to identify targetable cellular processes that modulate downstream inflammatory and catabolic activity in articular chondrocytes in response to a variety of stimuli.

Metabolic reprogramming is one such process that has gained interest in various cell types and disease state as a disease driver and may also be important in OA (*Chiellini, 2020*; *Zheng et al., 2021*). Intracellular metabolism does far more than energy production and can modulate the inflammatory response of cells through regulation at various levels, ranging from epigenetics modifications to redox modulation (*Gaber et al., 2017*; *Lu and Wang, 2018*). As such, chondrocyte intracellular metabolism has come into focus recently as a potential therapeutic target for modulating catabolic activity through regulation of inflammatory signaling pathways. Supporting this finding, we have shown recently that inflammatory stimuli alter the metabolism of chondrocyte, which can then regulate inflammatory responses (*Arra et al., 2020*).

Glucose metabolism has been fairly extensively studied in chondrocytes, though the role of other substrate pathways, such as fatty acid or amino acid metabolism, has been less well studied. However, understanding the role of these other substrate pathways is critical since many metabolic pathways are interconnected and likely play a role in OA pathogenesis. In support of this claim, several groups have displayed recently that modulation of metabolic pathways can protect against OA and rheumatoid arthritis (RA) in animal models (*Abboud et al., 2018*; *Coleman et al., 2018*; *Liu-Bryan, 2015*; *Ohashi et al., 2021*; *Shen et al., 2019*). In humans, studies have shown that OA and RA joints have altered metabolite levels in the synovial fluid, though it is unclear if these changes are due to disease or drivers of disease (*Akhbari et al., 2020*; *Kim et al., 2014*; *Zhai, 2019*; *Zheng et al., 2017*). Finally, it has been displayed that systemic metabolic diseases such as obesity, diabetes, and hypercholesterolemia (*Baudart et al., 2017*) likely influence OA development, potentially through nutrient availability (*Sellam and Berenbaum, 2013*; *Zhuo et al., 2012*). Based on these findings, it is probable that altered cell metabolism can be used not only as a biomarker of joint health but also as a therapeutic target. To help address some of the knowledge gaps in the field, we focused in this study on the role of glutamine in chondrocyte physiology and in response to inflammatory stimulation.

Glutamine is highly abundant throughout the body and is essential for many anabolic processes but can also be utilized for energy production (*Cruzat et al., 2018*). Some recent studies have elaborated the function of glutamine metabolism in chondrocyte physiology, though more understanding is required. One group demonstrated that inflammatory stimulation altered glutamine uptake and glutamate release in chondrocytes, with glutamate receptor involved in modulation of chondrocytes inflammatory response (*Piepoli et al., 2009*). Another group showed that glutamine metabolism is critical for regulating anabolic activity, glutathione production, and epigenetic modifications in chondrocytes (*Stegen et al., 2020*). Furthermore, several studies have highlighted that glutamine levels are altered in synovial fluid of OA patients (*Akhbari et al., 2020*; *Anderson et al., 2018*). These

studies highlight that glutamine is likely to be an important substrate for chondrocytes, both in healthy and disease states.

In this study, we aim to characterize the role of glutamine metabolism in the inflammatory response of chondrocytes. Recent work has begun to explore the role of glutamine in the chondrocyte inflammatory response, primarily utilizing glutamine supplementation to display potentially protective effects (*Ma et al., 2022*). However, the overall paucity of data in the field suggests the need for further research and a different approach to understanding the role of glutamine in OA chondrocytes. We utilize glutamine deprivation in the context of IL-1β stimulation as well as supplementation of glutamine downstream products to determine their function in the inflammatory response. We focus on NF-κB signaling as well as autophagy, both of which have been shown to be cooperatively important players in OA disease.

## Results

### Chondrocytes utilize glutamine for intracellular energy metabolism

We have previously shown that chondrocytes under inflammatory conditions undergo metabolic reprogramming, with increased reliance upon glycolysis and decreased oxidative phosphorylation (*Arra et al., 2020*). Several studies have demonstrated that there is mitochondrial dysfunction with inflammatory stimulation (*Arra et al., 2020*; *López-Armada et al., 2006*). During these conditions, there is decreased reliance upon glucose as a source of TCA cycle substrates, though other energy substrates such as glutamine can still fuel TCA activity to drive anabolic reactions (*Martínez-Reyes and Chandel, 2020*; *Meiser et al., 2016*). Thus, we seek to determine if chondrocytes can utilize glutamine to fuel anaplerotic TCA cycle activity. We observe that IL-1β stimulation alters the expression of glutamine and glutamate transports, as well as several key glutamine metabolic enzymes (*Figure 1A–E*).

However, enzyme expression may not necessarily reflect glutamine utilization. To determine this, we measured viability of chondrocytes in the presence or absence of glutamine. Culturing chondrocytes in glutamine free media led to a slight decrease in viability, suggesting that chondrocytes require glutamine for energy production (*Figure 1F*). Since the first step of glutamine metabolism and entry to TCA cycles is conversion to glutamate, a process catalyzed by glutaminase (GLS), we utilized a GLS inhibitor, CB-839, which mimics the effect of glutamine deprivation on cell metabolism by preventing glutamate generation. We confirmed that GLS inhibition significantly reduced intracellular glutamate to levels approaching that of glutamine deprivation (*Figure 1G*). We also observed a decrease in ATP levels with GLS inhibition, further confirming that chondrocytes do utilize glutamine for energy production (*Figure 1H*). Given that chondrocytes clearly rely upon glutamine, we then performed Seahorse analysis to confirm that chondrocytes utilized glutamine for metabolism and energy production. We note that IL-1β stimulation increases glycolysis and causes a dramatic decrease in OxPhos, likely via mitochondrial dysfunction (*Figure 1I–L*). We observed that glutamine deprivation led to a decrease in both extracellular acidification rate (ECAR) and oxygen consumption rate (OCR), with a more dramatic effect on OCR, likely due to contribution of glutamine to TCA anaplerotic activity (*Figure 1I–L*). ECAR is generally viewed as a surrogate for glycolytic activity in the form of lactic acid mediated acidification, though there is also a contribution of carbon dioxide generated from de-carboxylation during TCA cycle. We observe similar effects on metabolism with GLS inhibition, indicating that glutamine breakdown by GLS is an essential step for energy metabolism (*Figure 1—figure supplement 1A-D*).

We sought to determine if human OA cartilage also exhibits altered expression of glutamine metabolic enzymes. We noted that human OA cartilage displays increased expression of various glutamine metabolic enzymes (*Figure 1—figure supplement 1E-H*), potentially suggesting increased glutamine metabolism. We validate that OA cartilage is more catabolic and inflammatory through measurement of NF-κB inhibitor zeta (*Nfkbiz*) and matrix metalloprotease 3 (*Mmp3*) expression, respectively (*Figure 1—figure supplement 1I-J*). We also note that IL-1β stimulation of human chondrocytes isolated from knee cartilage caused some glutamine metabolic enzyme changes, though less significantly than OA chondrocytes (*Figure 1—figure supplement 1K-M*), indicating that metabolic changes in human cells may be a chronic change or in response to other inflammatory stimuli.

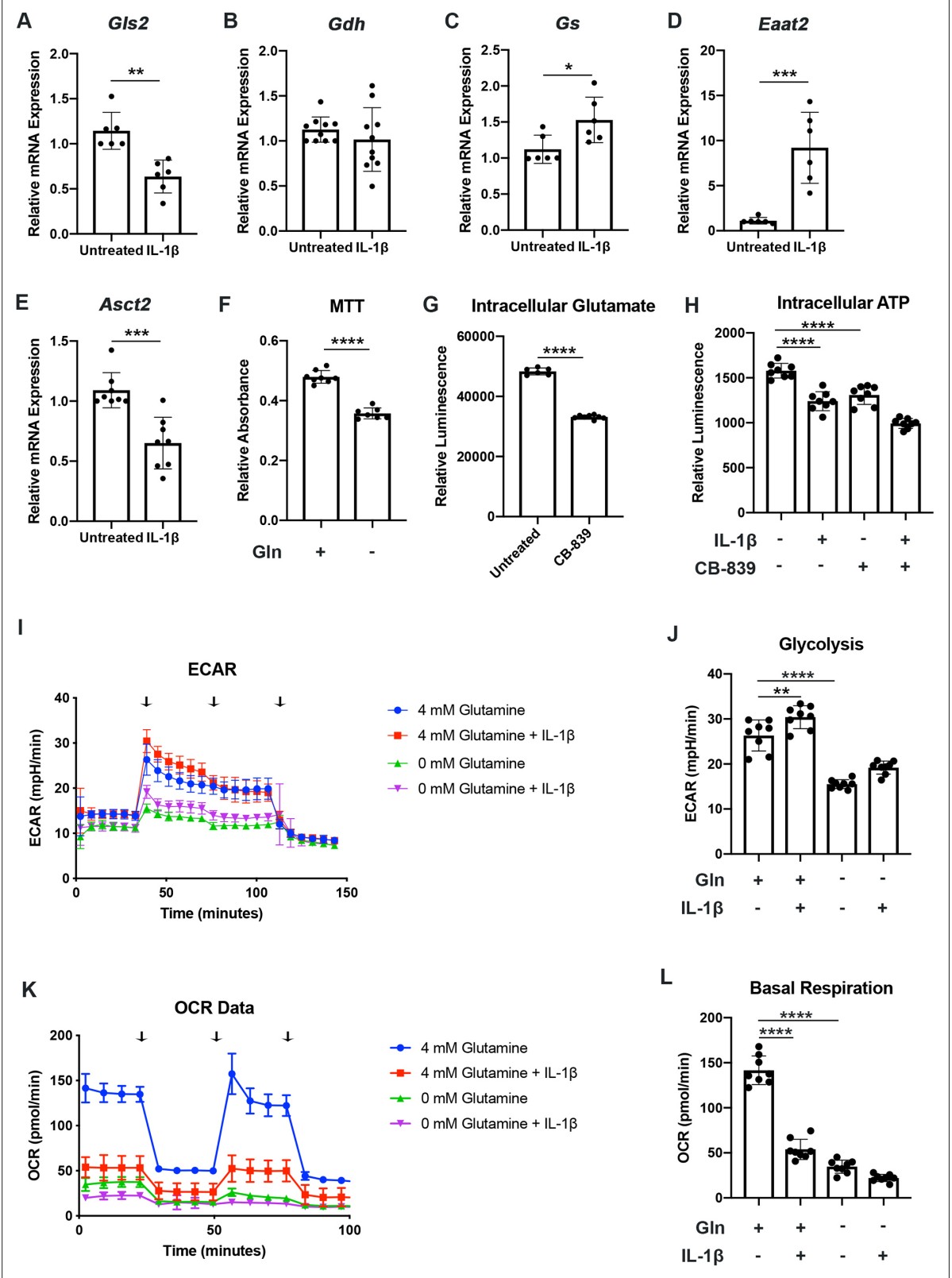

**Figure 1.** Chondrocytes rely upon glutamine for energy production. (**A–E**) Primary murine chondrocytes were treated with IL-1β (10ng/mL) for 24hr. Gene expression of *Gls2*, *Gdh*, *Gs*, *Eaat2*, and *Asct2* was measured by quantitative PCR. Results from n=6-10 independent biological samples (as depicted in individual panels). Unpaired Student's T test was performed (A: **p=0.0011, B: p=0.375, C: *p=0.0227, D: ***p=0.0005, and E: ***p=0.0003). (**F**) Primary murine chondrocytes were cultured in media with 4mM glutamine and 0mM glutamine under constant glucose conditions. After 24hr,

*Figure 1 continued on next page*

*Figure 1 continued*

viability was measured by (3-(4, 5-dimethylthiazolyl-2)-2, 5-diphenyltetrazolium bromide) (MTT) assay. Results from n=6 samples from one representative experiment. Unpaired Student's T test was performed, ****p<0.0001. (**G**) Primary murine chondrocytes were treated with CB-839 (1uM). Intracellular glutamate was measured by luminescent assay (n=6). Unpaired Student's T test was performed, ****p<0.0001. (**H**) Primary murine chondrocytes were treated with CB-839 and/or IL-1β for 24hr. Intracellular ATP was measured by luminescent assay. Results from one representative experiment (n=8). One-way ANOVA was performed followed by Tukey's multiple comparisons test, ****p<0.0001. (**I–L**) Primary sternal chondrocytes were cultured in media containing glutamine or media without glutamine for 24hr. Cells were then treated with IL-1β (10ng/mL) for 24hr. All values were normalized to cell viability of treatments relative to untreated cells as measured by MTT assay. (**I–J**) Extracellular acidification rate (ECAR) measurement in glycolysis stress test (Injection 1: no treatment, Injection 2: glucose, Injection 3: oligomycin, and Injection 4: 2-DG) or (**K–L**) Oxygen consumption rate (OCR) measurement in MitoStress test (Injection 1: no treatment, Injection 2: oligomycin, Injection 3: Carbonyl cyanide-p-trifluoromethoxyphenylhydrazone (FCCP), and Injection 4: antimycin A/rotenone) was performed on Seahorse Instrument. Measurements were performed every 6min with n=eight replicates per timepoint for each condition. Arrows represent injections timepoints. Graphs shown in *Figure 1J and L* are from a single timepoint. One-way ANOVA was performed followed by Tukey's multiple comparisons test. J:**p=0.0077 and ****p<0.0001; L:****p<0.0001.

The online version of this article includes the following source data and figure supplement(s) for figure 1:

**Source data 1.** Depicting original raw data related to *Figure 1*.

**Figure supplement 1.** Chondrocytes rely upon glutamine for energy production.

**Figure supplement 1—source data 1.** Depicting original raw data related to *Figure 1—figure supplement 1*.

## Glutamine deprivation causes metabolic reprogramming to inhibit glycolysis

Given that glutamine primarily supplies TCA cycle activity (*Yoo et al., 2020*), and not glycolytic substrates, we were surprised to observe that glutamine deprivation of chondrocytes was able to cause a reduction in both glycolysis and oxidative phosphorylation. We sought to determine what impact glutamine deprivation may have on systems such as glycolysis which tend to primarily utilize glucose, and if glutamine modulation is able to modify glycolytic activity. Furthermore, we have previously shown that metabolic reprogramming toward increased glycolysis induced by IL-1β can promote catabolic activity and OA disease (*Arra et al., 2020*).

We observe that glutamine deprivation itself was able to induce metabolic reprogramming that supports TCA activity. We noted increased expression of glutaminase (*Gls*) with glutamine deprivation (*Figure 2—figure supplement 1A*) and slight increase in some TCA cycles enzyme expression such as malate dehydrogenase (*Mdh*) and succinate dehydrogenase subunit A (*Sdha*), with insignificant changes in others (*Figure 2—figure supplement 1B-E*). However, glutamine deprivation inhibited the expression of various glycolytic and pentose phosphate pathway (PPP) enzymes (*Figure 2A–C*). Furthermore, we observe that glutamine deprivation can prevent many of the glycolytic and PPP enzymes metabolic changes observed with IL-1β stimulation (*Figure 2A–C*). This is further supported by the finding that glutamine deprivation can reduce lactate production by chondrocytes, a marker of glycolytic activity (*Figure 2—figure supplement 1F*), confirming that the decrease in ECAR seen with glutamine deprivation is at least partially due to decreased glycolytic activity (*Figure 1J*). Based on our findings, it appears that glutamine deprivation supports TCA cycle activity but inhibits glycolysis and PPP. This may be a compensatory mechanism utilized by cells in the absence of glutamine to sustain ATP production via utilization of the energy favorable TCA cycle, as well as anabolic activity.

We then performed some targeted proteomics in the context of glutamine deprivation (*Figure 2D–E*) and noted that glutamine deprivation does not reduce levels of TCA metabolites such as α-KG, malate, and oxaloacetate but does reduce levels of pyruvate generated from glycolysis (*Figure 2D–K*). However, we did note that glutamine is in fact required for the production of various downstream substrates, such as asparagine and aspartate (*Figure 2I–J*). Given that glutamine deprivation reduced OxPhos activity and ATP levels, but did not reduce TCA metabolite levels, chondrocytes may be able to generate TCA metabolites by utilization of other anaplerotic processes. These systems can generate metabolites but usually do not generate energy. As an example, we noted increased expression of *Psat1* with glutamine deprivation (*Figure 2—figure supplement 1G*), which has recently been displayed to be one source of glucose-based α-KG to fuel TCA cycle (*Hwang et al., 2016*).

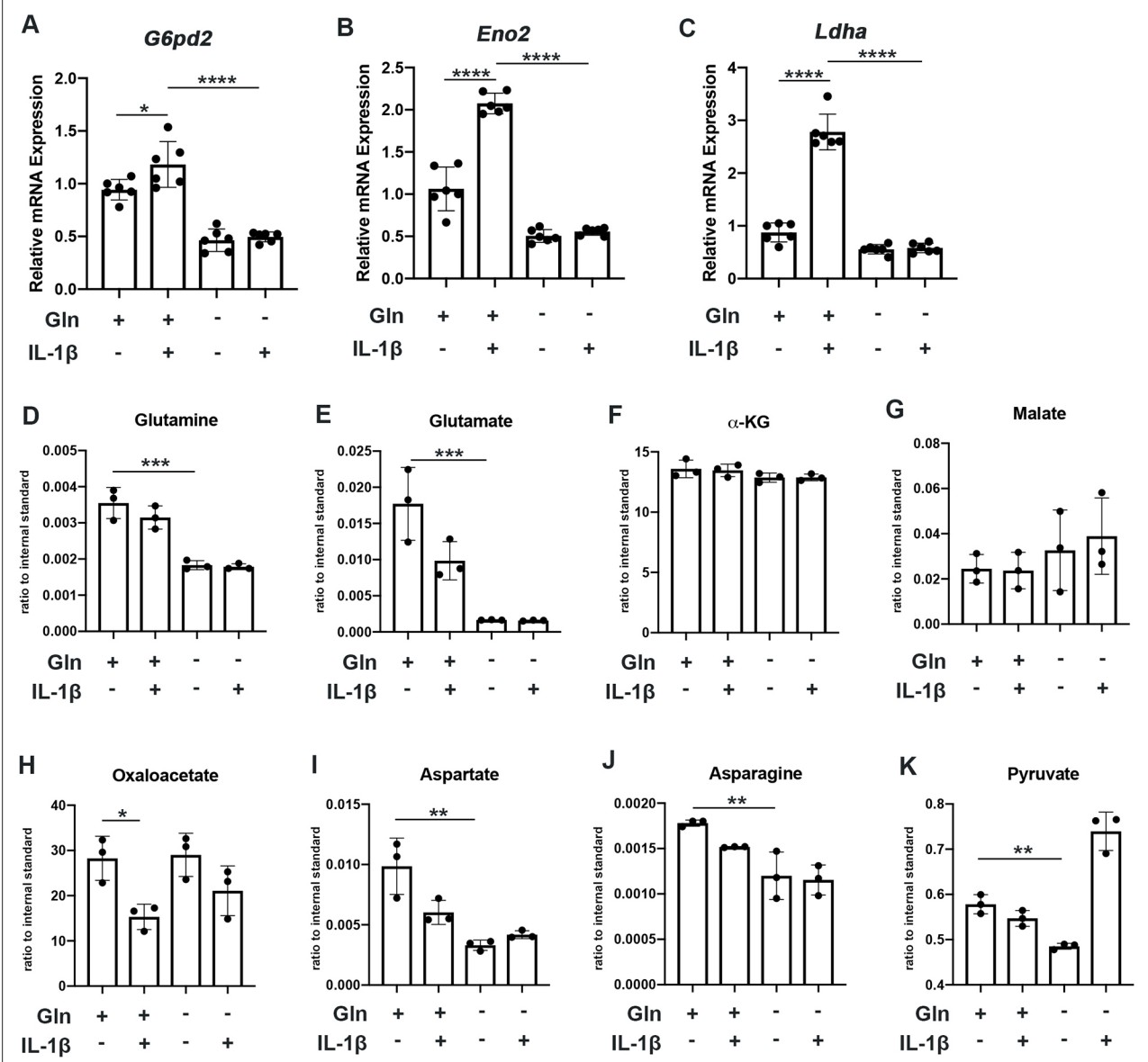

**Figure 2.** Glutamine deprivation causes metabolic reprogramming to inhibit glycolysis. (**A–C**) Primary murine chondrocytes were cultured in media containing 4 mM glutamine or 0 mM glutamine for 24 hr. Cells were then treated with IL-1β (10 ng/mL) for 24 hr. Gene expression of *G6pd2*, *Eno1*, and *Ldha* was measured by quantitative PCR from n=6 replicates. One-way ANOVA was performed followed by Tukey's multiple comparisons test. A: *p=0.0242, ****p<0.0001, B: ****p<0.0001, and C: ****p<0.0001. (**D–K**) Under similar conditions, metabolite levels were measured by Liquid chromatography–mass spectrometry (LC-MS) with n=3 replicates. One-way ANOVA was performed followed by Tukey's multiple comparisons test. D: ***p=0.0003, E:***p=0.0006, F: p>0.05, G: p>0.05, H:*p=0.036, I: **p=0.0012, J:**p=0.0079, and K: **p=0.009.

The online version of this article includes the following source data and figure supplement(s) for figure 2:

**Source data 1.** Depicting original raw data related to *Figure 2*.

**Figure supplement 1.** Glutamine deprivation causes metabolic reprogramming to inhibit glycolysis.

**Figure supplement 1—source data 1.** Depicting original raw data related to *Figure 2—figure supplement 1*.

## Glutamine metabolism by GLS contributes to inflammatory gene expression

We have previously depicted that altered metabolism can regulate the inflammatory and catabolic response of chondrocytes. Since the metabolic changes induced by glutamine deprivation opposed

the metabolic changes we observe with IL-1β stimulation, we suspected that glutamine may also play a role in modulating the inflammatory response induced by IL-1β.

To determine the role of glutamine metabolism in the inflammatory response, we cultured chondrocytes in media with and without glutamine under sufficient glucose conditions and treated them with IL-1β. We observed that glutamine deprivation led to a decrease in inflammatory and catabolic gene expression in response to IL-1β, with a reduction in expression of genes such as interleukin 6 (*Il6*) and matrix metalloprotease 13 (*Mmp13*) (*Figure 3A–B*). We then sought to determine the mechanism by which glutamine can regulate the inflammatory response. We measured NF-κB activity since it is the principle inflammatory response pathway to IL-1β that we have previously demonstrated is important for OA development (*Arra et al., 2022*; *Arra et al., 2020*). We observe using chondrocytes derived from p65-luciferase reporter mice that glutamine deprivation dose dependently inhibits NF-κB activation, as measured by luciferase activity (*Figure 3C*). It has also previously been displayed that IκB-ζ is a critical pro-inflammatory mediator of NF-κB activity in chondrocytes treated with IL-1β (*Arra et al., 2022*; *Choi et al., 2018*). We observe that glutamine deprivation leads to a decrease in IκB-ζ protein expression and stability, at least partially due to inhibition of NF-κB activity (*Figure 3D*). Our earlier work has also shown that IκB-ζ is a redox sensitive protein that is stabilized by oxidative stressors from metabolic sources such as Lactate Dehydrogenase A (LDHA) and the mitochondria, so we measured reactive oxygen species (ROS) levels in the absence of glutamine in response to IL-1β stimulation. We observed decreased ROS production in the absence of glutamine (*Figure 3E*), likely due to decreased inflammatory response and a reduction in pro-oxidative metabolic changes we have previously characterized. It should be noted that glutamine is important for the production of the antioxidant molecule glutathione in chondrocytes; however, this result seems to suggest that glutamine deprivation can reduce oxidative species generation more significantly than glutathione production may be reduced (*Stegen et al., 2020*).

We then interrogated if GLS inhibition can modulate the inflammatory response similar to glutamine deprivation. We observed that GLS inhibition by CB-839 was also effective at decreasing the inflammatory response, suggesting that glutamine to glutamate conversion is important for the inflammatory response (*Figure 3—figure supplement 1A*). We also observed that GLS inhibition was also able to potently decrease IκB-ζ protein expression (*Figure 3—figure supplement 1B*), indicating that glutaminolysis contributes to IκB-ζ-mediated gene expression. Furthermore, GLS inhibition reduced NF-κB activation (*Figure 3—figure supplement 1C*).

The glutaminolysis reaction performed by GLS generates glutamate and free ammonia from glutamine (*Yoo et al., 2020*). Glutamate can then be converted to a-ketoglutarate, glutathione, or undergo transaminase reactions. On the other hand, ammonia is a reactive species, often viewed as a waste product, and can be incorporated into amino acids or urea for its removal (*Kurmi and Haigis, 2020*; *Spinelli et al., 2017*). We sought to determine if glutamate or ammonia generated by GLS can regulate the inflammatory response of chondrocytes. We supplemented chondrocytes with ammonia or glutamate and measured the inflammatory response. We observed that ammonia supplementation was pro-inflammatory, activating NF-κB and increasing IκB-ζ protein levels in the setting of IL-1β stimulation (*Figure 3—figure supplement 1D*). It also increased expression of inflammatory and catabolic genes (*Figure 3G–H*). Ammonia supplementation also partially rescued inflammatory gene expression under glutamine deprivation conditions. Glutamate supplementation did not have a significant impact on NF-κB activation (*Figure 3—figure supplement 1D*). This finding suggests that ammonia generation from glutamine metabolism may be involved in promoting inflammation through stabilization of IκB-ζ and its transcriptional program. While glutamine is a major source of ammonia production, it is not the only source of ammonia, as asparagine is another amino acid with an amide side chain that can generate ammonia. We then sought to determine if asparagine supplementation in the absence of glutamine may be able to rescue the effect of glutamine deprivation. However, we observed that asparagine supplementation was unable to do so, suggesting that glutamine deprivation may activate unique pathways (*Figure 3—figure supplement 1E*).

Given that we did not observe an increase in inflammation with glutamate supplementation, we then tested the efficacy of glutamate dehydrogenase (GDH) inhibitor, EGCG, which blocks the conversion of glutamate to α-KG (*Li et al., 2006*), a process that also releases an ammonia group and is critical for the generation of pro-inflammatory downstream metabolites such as succinate. We note that EGCG treatment also slightly reduced the inflammatory response represented by *Il6* expression,

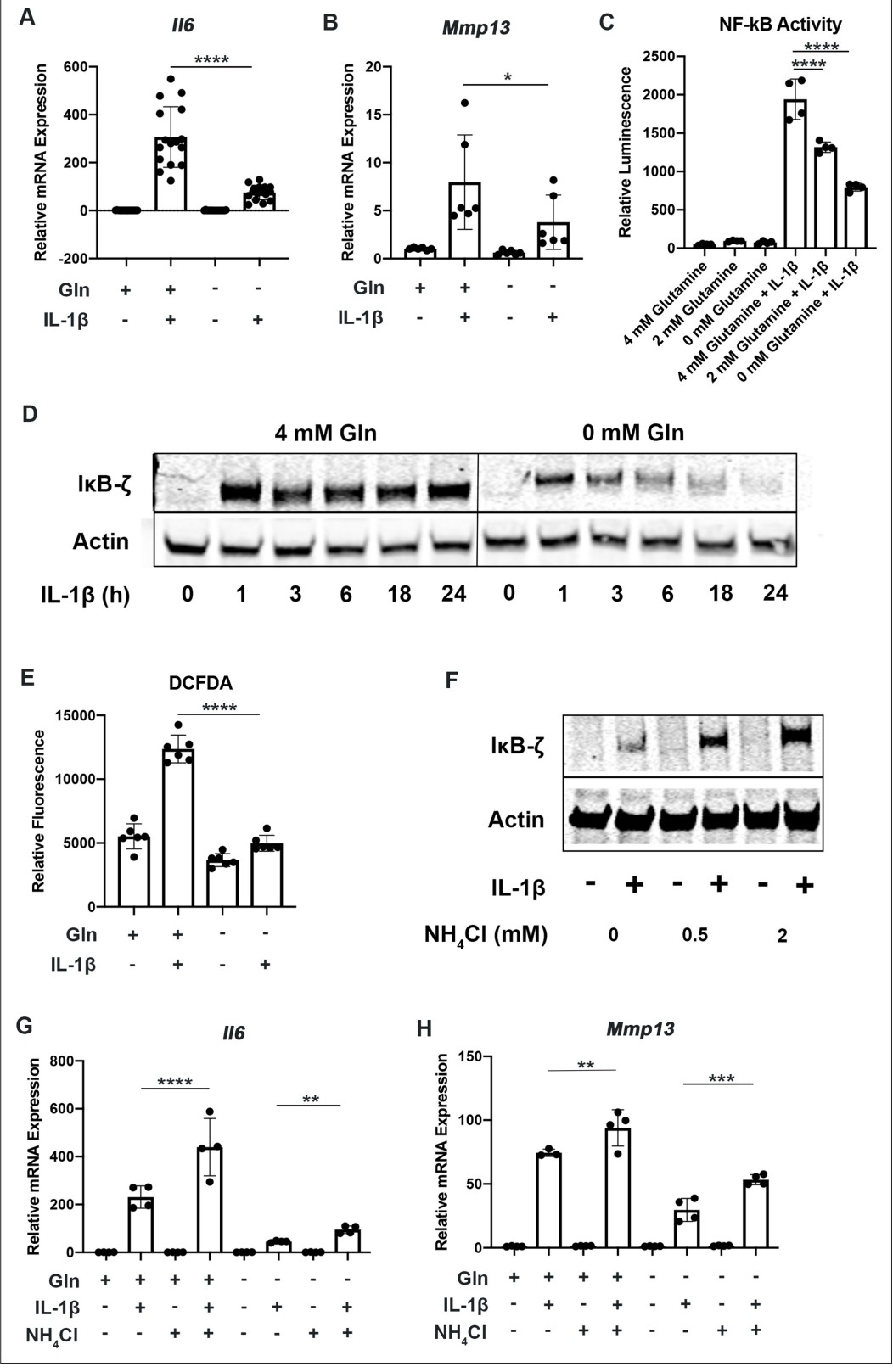

**Figure 3.** Glutamine deprivation inhibits the inflammatory response. (**A–B**) Primary murine chondrocytes were cultured in media containing 4 mM glutamine or 0 mM glutamine for 24 hr. Cells were then treated with IL-1β (10 ng/mL) for 24 hr. Gene expression of *Il6* and *Mmp13* was measured by quantitative PCR (qPCR). One-way ANOVA was performed followed by Tukey's multiple comparisons test. A:****p<0.0001 (n=16) and B: *p=0.0489 (n=6).

*Figure 3 continued on next page*

*Figure 3 continued*

(**C**) Primary murine chondrocytes were isolated from NF-$\kappa$B-luciferase reporter mice. Chondrocytes were then cultured in media containing 4 mM, 2 mM, or 0 mM glutamine for 24 hr. Cells were then treated with IL-1β for 24 hr. NF-$\kappa$B activity was measured by luciferase assay. n=4. One-way ANOVA was performed followed by Tukey's multiple comparisons test. ****p<0.0001. (**D**) Primary murine chondrocytes were cultured in media containing 4 mM glutamine or 0 mM glutamine for 24 hr. Cells were treated with IL-1β for the indicated timepoints. I$\kappa$B-$\zeta$ protein (85kDa) was measured by immunoblotting, with actin (42kDa) used as housekeeping. Image displays representative experiment. (**E**) Primary murine chondrocytes were cultured in media containing 4mM glutamine or 0 mM glutamine for 24 hr. Cells were then treated with IL-1β (10 ng/mL) for 24 hr. ROS levels were measured by 2',7'–dichlorofluorescin diacetate (DCFDA) assay using microplate reader. n=6. One-way ANOVA was performed followed by Tukey's multiple comparisons test. ****p<0.0001. (**F**) Primary chondrocytes were cultured in media containing glutamine and supplemented with ammonium chloride at the indicated concentrations for 24 hr in the presence of IL-1β. I$\kappa$B-$\zeta$ protein was measured by immunoblotting. (**G–H**) Primary chondrocytes were cultured in media containing 4 mM or 0 mM glutamine for 6 hr. Cells were then supplemented with or without 2 mM ammonium chloride. IL-1β stimulation was performed for 24 hr. Gene expression of *Il6* and *Mmp13* was measured by qPCR. n=4. One-way ANOVA was performed followed by Tukey's multiple comparisons test. G: ****p<0.0001, **p=0.0065, and H: **p=0.0096, ***p=0.0005.

The online version of this article includes the following source data and figure supplement(s) for figure 3:

**Source data 1.** Depicting original raw data related to *Figure 3*.

**Source data 2.** Original raw data for *Figure 3D, F*.

**Figure supplement 1.** Glutamine deprivation inhibits the inflammatory response.

**Figure supplement 1—source data 1.** Depicting original raw data related to *Figure 3—figure supplement 1*.

**Figure supplement 1—source data 2.** Original raw data related to *Figure 3—figure supplement 1B*.

though less potently than glutamine deprivation (*Figure 3—figure supplement 1F*). This is expected given that EGCG blocks α-KG production and glutamate-based ammonia release in TCA cycle but does not alter glutamine generation into glutamate or other downstream products. Hence, EGCG only partially mimics glutamine deprivation and as such is less anti-inflammatory.

## Glutamine deprivation activates autophagy

Since it is well known that nutrient deprivation can induce autophagy (*Russell et al., 2014*), we sought to determine what impact glutamine deprivation would have on chondrocyte autophagy processes, especially in the context of inflammation. We note that IL-1β stimulation of chondrocytes leads to a decrease in autophagy, as noted by accumulation of p62 protein. We then observe that there is an upregulation of autophagy with glutamine deprivation, as indicated by a decrease in p62 protein, which is often an indication of autophagy progression as p62 is degraded by autophagosomes (*Figure 4A*, quantified in *Figure 4—figure supplement 1A-B*). We also note a significant decrease in LC3 protein levels with glutamine deprivation due to increased consumption of LC3 through autophagy. We validated this through chloroquine treatment, an inhibitor of autophagy, which can rescue LC3 and p62 levels in glutamine deprivation conditions, indicating that LC3 and p62 are being processed by autophagy. We validated these findings by immunofluorescence, which displayed that chloroquine treatment led to far greater increase in LC3-positive punctate in cells under glutamine deprivation conditions compared to glutamine replete conditions (*Figure 4B*). We confirm that these findings are due to protein processing since we do not observe similar changes at the gene expression level (*Figure 4—figure supplement 1C-D*). Interestingly, we note that glutamine deprivation also leads to a transient decrease in gene expression of microtubule-associated protein 1A light chain 3b (*Lc3b*) and sequestosome (*Sqstm1*) at less than 24 hr, which recovers at the 24 hr timepoint (*Figure 4—figure supplement 1E-F*). We also observe that LC3 protein levels start to decrease rapidly with glutamine deprivation, but p62 levels do not decrease until 48 hr (*Figure 4—figure supplement 1G*). We observed similar effects with GLS inhibition by CB-839 (*Figure 4—figure supplement 1H*).

We then supplemented chondrocytes deprived of glutamine with glutamate or ammonia and measured levels of LC3 and p62 to determine how glutaminolysis affects autophagy. We noted ammonia supplementation was able to inhibit autophagy and reverse the effect of glutamine deprivation on LC3 and p62 expression, similar to chloroquine treatment (*Figure 4C*, Quantified in *Figure 4—figure supplement 1J-K*, *Figure 4—figure supplement 1I*). Ammonia treatment led to an increase

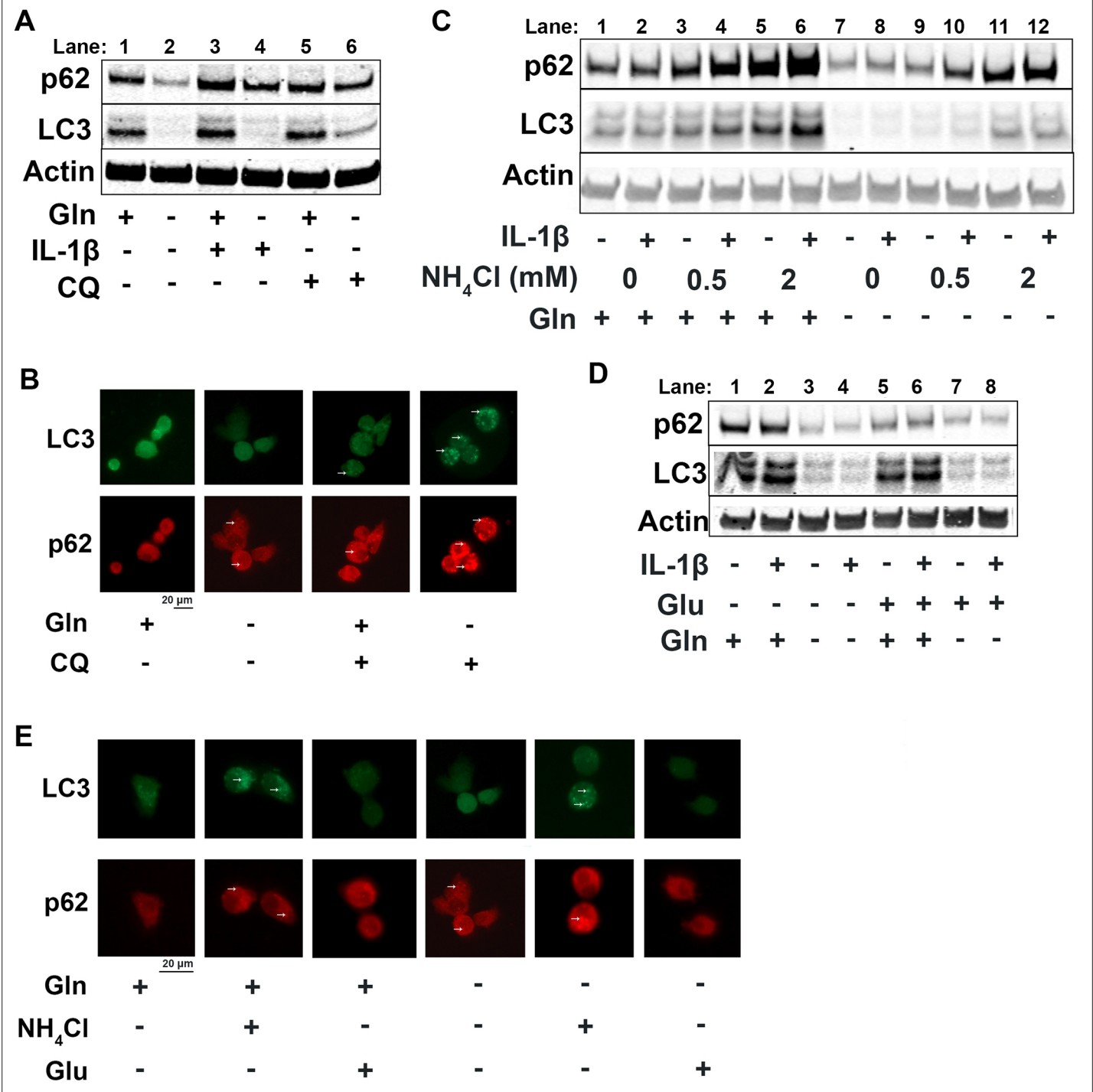

**Figure 4.** Glutamine deprivation promotes autophagy, and ammonia inhibits autophagy. (**A**) Primary murine chondrocytes were cultured in media containing 4 mM glutamine or 0 mM glutamine for 24 hr. Cells were then treated with IL-1β (10 ng/mL) in the presence or absence of chloroquine (10 μM) for 24 hr. Protein expressions of p62 (62kDa) and LC3-II (17kDa) were measured by immunoblotting, with representative image displayed. Bands quantified in supplemental figure. (**B**) Primary murine chondrocytes were plated on coated cover slips cultured in glutamine containing or glutamine free media for 12 hr. Cells were treated with chloroquine (10 μM) for 6 hr. Cells were fixed with 4% formaldehyde in PBS, and immunofluorescence (IF) was performed for LC3B and p62. Cells were mounted on slides and imaged with representative images displayed. (**C**) Primary chondrocytes were cultured in media containing 4 mM or 0 mM glutamine. Cells were supplemented with ammonium chloride at the indicated concentrations. After 6 hr, cells were treated with IL-1β (10 ng/mL) for 24 hr. Immunoblotting was performed for p62 and LC3B to display autophagosome processing. Image displays representative experiment. Bands quantified in supplemental figure. (**D**) Primary chondrocytes were cultured in media containing 4 mM or 0 mM glutamine. Cells were supplemented with glutamate (200 μM). After 6 hr, cells were treated with IL-1β (10 ng/mL) for 24 hr. Immunoblotting was

*Figure 4 continued on next page*

*Figure 4 continued*

performed for p62 and LC3b. Image displays representative experiment. Bands quantified in supplemental figure. (**E**) Primary murine chondrocytes were plated on coated cover slips cultured in glutamine containing or glutamine free media for 12 hr. Cells were supplemented with ammonium chloride (2 mM) or glutamate (200 μM). Cells were fixed with 4% formaldehyde, and IF was performed for LC3b and p62. Cells were mounted on slides and imaged with representative images to display autophagosome punctate.

The online version of this article includes the following source data and figure supplement(s) for figure 4:

**Source data 1.** Depicting original raw data related to *Figure 4*.

**Source data 2.** Original raw data related to *Figure 4A,C,D*.

**Figure supplement 1.** Glutamine deprivation promotes autophagy, and ammonia inhibits autophagy.

**Figure supplement 1—source data 1.** Depicting original raw data related to *Figure 4—figure supplement 1*.

**Figure supplement 1—source data 2.** Original raw data related to *Figure 4—figure supplement 1G,H,I,O*.

in LC3b and p62, likely through blockade of autophageosome-lysosome fusion. We also observe that glutamate supplementation appeared to increase autophagy, as noted by a decrease in p62 levels (*Figure 4D*, quantified in *Figure 4—figure supplement 1L-M*, *Figure 4—figure supplement 1I*). These findings were confirmed through increased number of LC3 positive punctate in ammonia treated cells but not in glutamate treated cells, indicating opposing effects of ammonia and glutamate on autophagy (*Figure 4E*).

One of the major cell stress response factors involved in regulating metabolism and autophagy is activating transcription factor 4 (ATF4), which can modulate intracellular metabolism (*B'chir et al., 2013*; *O'Leary et al., 2020*; *Stegen et al., 2022*). Mechanistically, we observe that glutamine deprivation is able to increase ATF4 protein expression, which is reduced with IL-1β stimulation (*Figure 4—figure supplement 1N-O*). We note that at around 12 hr of glutamine deprivation, *Atf4* expression increases (*Figure 4—figure supplement 1P*). ATF4 activation is a well-known response system to amino acid deprivation, and it is known to be a driver of autophagy processes (*Jin et al., 2021*; *Ye et al., 2010*). ATF4 is also a mediator of metabolic reprogramming, which we observe with glutamine deprivation and IL-1β stimulation. Suspecting that ATF4 may be important in OA, we note that OA mouse cartilage (meniscal-ligamentous injury [MLI]) has decreased expression of ATF4, mimicking the effect of IL-1β stimulation (*Figure 4—figure supplement 1Q*). Based on these results, it is predicted that ATF4 may be an anti-inflammatory factor and will be the focus of future work.

## mTOR2 but not mTOR1 is activated by glutamine deprivation

Since mammalian target of rapamycin (mTOR) signaling is another critical factor connecting metabolism and autophagy, we interrogated mTOR activation in the setting of glutamine deprivation and inflammation. mTOR activation has been shown to be a driver of metabolic changes, especially for glycolytic pathways (*Linke et al., 2017*; *Magaway et al., 2019*). Glutamine deprivation decreased glycolysis and glycolytic enzyme expression, hence we suspected that mTOR modulation may be involved. We note that mTOR1 and mTOR2 activity are increased with IL-1β stimulation as measured by increased phosphorylation of S6 riboprotein and phosphorylation of AKT-473 (*Figure 5A–B*), supporting the inhibition in autophagy we previously measured. With glutamine deprivation, we noticed a decrease in mTOR1 activity, as measured by phosphorylation of S6 ribosomal protein but an increase in mTOR2 activity, as measured by increased AKT-473 phosphorylation (*Figure 5A–B*).

We then utilized rapamycin, an mTOR1 inhibitor, to interrogate the role of mTOR in the inflammatory response and glutamine metabolic gene expression changes. Rapamycin has been suspected to be protective in many disease states through the downstream effects of mTOR inhibition, such as increasing longevity and preventing aging-associated diseases in animal models (*Selvarani et al., 2021*). We validate that rapamycin can block mTOR1 activation through complete abrogation of phospho-S6 expression (*Figure 5—figure supplement 1A*). We show that rapamycin treatment can reverse some metabolic changes induced by IL-1β, such as increased glycolytic enzyme expression using LDHA as a representative gene (*Figure 5—figure supplement 1B*). We also note that rapamycin treatment is able to upregulate expression of glutamine synthase (*Gs*) and *Gls* but did not affect glutamic-oxaloacetic transaminase 2 (*Got2*) expression (*Figure 5—figure supplement 1C-E*). However, under glutamine deprivation conditions, rapamycin treatment only affected *Gls* expression, but not *Gs* or *Got2*. We then measured the impact of mTOR1 inhibition by rapamycin on the inflammatory response

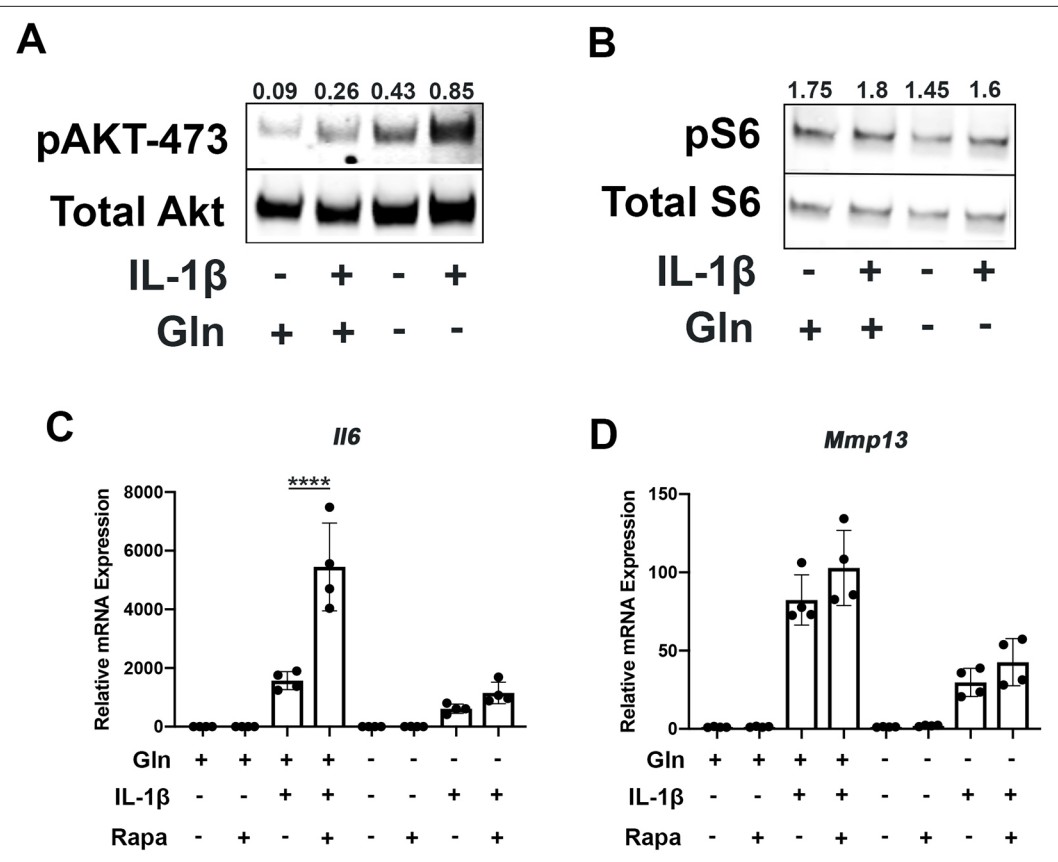

**Figure 5.** Glutamine deprivation modulates mTOR activation. (**A**) Primary murine chondrocytes were cultured in media containing 4 mM glutamine or 0 mM glutamine for 24 hr. Cells were then treated with IL-1β (10 ng/mL). After 24 hr, lysates were collected, and immunoblotting was performed for pAKT (60kDa) and total Akt (58kDa). (**B**) Under similar conditions, immunoblotting was performed for pS6 and total S6 (32kDa). (**C–D**) Primary murine chondrocytes were cultured in media containing 4 mM glutamine or 0 mM glutamine for 24 hr. Cells were then treated with IL-1β (10 ng/mL) in the presence or absence of rapamycin 50 nM for 24 hr. Gene expression of *Il6* and *Mmp13* was measured by quantitative PCR. Results from one representative experiment. n=4. One-way ANOVA was performed followed by Tukey's multiple comparisons test. C:****$p<0.0001$.

The online version of this article includes the following source data and figure supplement(s) for figure 5:

**Source data 1.** Depicting original raw data related to *Figure 5*.

**Source data 2.** Original raw data related to *Figure 5A, B*.

**Figure supplement 1.** Glutamine deprivation modulates mTOR activation.

**Figure supplement 1—source data 1.** Depicting original raw data related to *Figure 5—figure supplement 1*.

**Figure supplement 1—source data 2.** Original raw data related to *Figure 5—figure supplement 1A*.

---

and unexpectedly observed that rapamycin increases the expression of inflammatory and catabolic genes (*Figure 5C–D*). It also reverses some of the inflammatory inhibition induced by glutamine deprivation, suggesting an important role for mTOR1 in the anti-inflammatory effect of glutamine deprivation. Rapamycin treatment has been displayed systemically to be anti-osteoarthritic in several studies (*Pal et al., 2015*), though there is precedence for rapamycin to be pro-inflammatory in intra-cellular processes (*Weichhart et al., 2008*).

## Discussion

This work displays that glutamine utilization by chondrocytes is important for their physiology and inflammatory response and may be an important player in OA. We show that chondrocytes utilize glutamine for energy production even when glucose is abundant, indicating that chondrocytes rely

upon multiple substrates for metabolism. However, we observe that glutamine deprivation is able to decrease chondrocyte glycolysis and oxidative phosphorylation, which was unexpected since prior groups have mainly described glutamine as fueling the TCA cycle in the mitochondria (*Yoo et al., 2020*). This finding was enlightened when we noted that glutamine deprivation caused metabolic reprogramming to inhibit glycolytic activity by significantly decreasing expression of glycolytic enzymes but maintained TCA metabolites. This was an unanticipated new interaction between glutamine and glucose metabolism which may be a compensatory mechanism that forces chondrocytes to rely upon other energy sources such as fatty acid oxidation to maintain anabolic processes. In addition, this reprogramming caused by glutamine deprivation may actually be protective in the context of inflammation, as will be discussed further. Future work utilizing an untargeted metabolomics approach will be useful for understanding the contribution and interactions of various substrate pathways to chondrocyte metabolism and physiology.

From an inflammatory standpoint, which is important for OA pathophysiology and cartilage degradation, we observed that glutamine deprivation was able to decrease chondrocyte expression of inflammatory and catabolic genes in response to IL-1β stimulation. Mechanistically, we observed decreased NF-κB activation and expression of IκB-ζ with glutamine deprivation. We also note that glutamine deprivation reduces ROS generation in response to IL-1β stimulation, which we have previously shown is protective and can block IκB-ζ expression (*Arra et al., 2022*; *Arra et al., 2020*). This effect may be mediated via the metabolic reprogramming induced by glutamine deprivation, reversing metabolic changes induced by IL-1β that we have previously shown are pro-inflammatory. For example, LDHA and PPP activity have previously been shown to be pro-inflammatory in chondrocytes (*Arra et al., 2020*), but these processes are decreased in the setting of glutamine deprivation. In this manner, glutamine deprivation may further decrease the inflammatory response through a reduction in oxidative stressors and metabolic modulation. We find that our results may conflict with some recent work which showed that glutamine supplementation to chondrocytes is protective through reduction in inflammatory factors. There are several differences in the methodology used to study the impact of glutamine between these two studies. We utilize glutamine deprivation, while the other group utilized glutamine supplementation of up to 20 mM to media that already contained glutamine at 2.5 mM concentration. It is possible that both of these processes can activate protective mechanisms, such as autophagy or reduction in oxidative stress with deprivation, or production of α-KG and activation of long non-coding RNA (lncRNA's) with over-supplementation. Overall, it is clear that the role of substrate metabolism in chondrocytes requires more nuanced study.

We then noted that glutamine deprivation is able to promote autophagy and activates stress response systems such as ATF4 pathway. These systems are likely required for maintenance of amino acid levels and anabolic activity in the absence of glutamine. However, autophagy and stress systems have also been shown to be protective and may hold therapeutic potential (*Aman et al., 2021*). For example, intermittent fasting and rapamycin as autophagy promoting compounds have gained interest recently as modalities for driving protective autophagy (*Johnson et al., 2013*; *Mattson and de Cabo, 2020*). Our prior work has also demonstrated that autophagy can also regulate inflammatory responses through modulation of NF-κB and other pathways (*Adapala et al., 2020*), which may be involved in the anti-inflammatory effect of glutamine deprivation. Our future work will focus on understanding the role of ATF4 stress response system and autophagy in the modulation of chondrocyte inflammatory response and NF-κB activity. We can also utilize well-accepted autophagy assay systems such as LC3-GFP mice as well as knockout models of critical autophagy components in order to further study autophagic flux in the setting of nutrient deprivation. This can provide greater insight into the effects on autophagic flux and downstream impact on inflammatory processes.

Our work also briefly explored the interaction of mTOR and inflammatory response in the setting of nutrient deprivation. Interestingly, our work displays that inhibition of mTOR1 by rapamycin was pro-inflammatory and pro-catabolic, in contrast with other studies that showed that rapamycin is protective in vivo (*Caramés et al., 2012*; *Takayama et al., 2014*). This finding raises questions regarding the therapeutic potential of rapamycin for the treatment of aging associated disease such as OA, which has gained interest in recent the years (*Selvarani et al., 2021*). Rapamycin is well known to be an activator of autophagy, a supposedly 'protective' process, yet use of rapamycin increases the production of catabolic factors in the setting of inflammation. It is possible that these results are due to the complex interactions of mTOR1 and mTOR2, which can have significantly different downstream

functions (*Jhanwar-Uniyal et al., 2019*). It is possible that dysregulation of the balance of these systems can influence cell metabolism and also lead to a pro-inflammatory state (*Szwed et al., 2021*). Our future work will explore the differing roles of mTOR1 and mTOR2 in the chondrocyte inflammatory response to better understand the role of modulators such as rapamycin as therapeutics.

Our study then focused on the GLS reaction, which is one of the rate-limiting steps of glutamine metabolism (*Herranz, 2017*). We demonstrated that ammonia is an inhibitor of autophagy and promotes inflammatory responses, while glutamate is not. Prior studies on the role of ammonia in regulation of autophagy have demonstrated that ammonia is able to both induce and inhibit autophagy through various mechanisms (*Soria and Brunetti-Pierri, 2019*). It is possible that ammonia derived from glutamine may be pathological, especially if it is not appropriately recycled. The role of ammonia and ammonia-removal processes in chondrocytes require further study, especially in the context of inflammation, to determine their importance in joint disease. In addition, measurement of ammonia levels in synovial fluid may provide insight into the health of OA and RA joints. Our finding that glutamate treatment does not influence inflammatory response is in agreement with a prior study showing that exogenous glutamate did not influence inflammatory response of chondrocytes, although N-methyl-d-aspartate (NMDA) receptor blockade was anti-inflammatory, raising interesting questions about the role of glutamate in chondrocyte physiology (*Piepoli et al., 2009*). Overall, our work suggests that glutaminolysis may be pro-inflammatory through the production of ammonia which can block protective autophagy unless systems exist for ammonia incorporation and removal that can prevent this effect.

The results of this work lay the foundation for further investigation into glutamine metabolism as a possible therapeutic target. Several inhibitors exist that may hold some therapeutic potential, such as the GLS inhibitor CB-839, which is currently in clinical trials for anti-tumor potential and can be repurposed for the treatment of OA. Use of genetic mouse models will also provide more detailed in vivo pre-clinical information when combined with OA models. In addition, further work will be performed to determine if glutamine and downstream metabolite levels can be correlated to OA disease severity, allowing for the development of biomarkers. Another major knowledge gap is the understanding of non-glutamine amino acid and fatty acid utilization by chondrocytes, which can be performed through combined metabolomic, proteomic and transcriptomics-based approaches. Finally, more work needs to be performed using human samples such as articular cartilage and synovial fluid to create better translational models since OA is unlikely to be a single disease entity but a variety of sub-conditions with their own unique pathophysiology (*Deveza and Loeser, 2018*). A complete understanding of chondrocyte metabolism may provide an expanded toolbox for the understanding of OA and give rise to a personalized approach for patient treatment.

# Materials and methods

**Key resources table**

| Reagent type (species) or resource | Designation | Source or reference | Identifiers | Additional information |
|---|---|---|---|---|
| Genetic reagent (*Mus, musculus*) | C57BL/6mice | Jackson Labs | RRID:IMSR_JAX:000664 | |
| Genetic reagent (*Mus, musculus*) | NF-kB luciferase reporter mice | Jackson Labs | RRID:IMSR_JAX:027529 | |
| Biological sample (human) | Human osteoarthritis chondrocytes | Isolated from discarded human tissues, *Arra et al., 2020* | | |
| Biological sample (mouse) | Murine chondrocytes | Isolated from sterna of newborn pups of genetic strains indicated above, *Arra et al., 2020* | | |
| Antibody | Anti-ATF4 (rabbit polyclonal) | ThermoFisher | 10835–1-AP, RRID:AB_2058600 | 1:1,000 for Western blot |
| Antibody | Biotinylated secondary (horse anti-rabbit polyclonal) | Vector Biolabs | BP-1100 | 1:1,000 for IHC |
| Antibody | Anti-LC3b (rabbit polyclonal) | Cell Signaling Technology | 2775, RRID:AB_915950 | 1:1,000 Western blot; 1:100 for IF |

*Continued on next page*

*Continued*

| Reagent type (species) or resource | Designation | Source or reference | Identifiers | Additional information |
|---|---|---|---|---|
| Antibody | Anti-p62 (mouse monoclonal) | Abnova | 2C11, RRID:AB_437085 | 1:1,000 Western blot; 1:100 for IF |
| Antibody | Anti-IkB-z (rat monoclonal) | Invitrogen | 14-16801-82, RRID:AB_11218083 | 1:1,000 Western blot |
| Antibody | Anti-p-Akt (rabbit polyclonal) | Cell Signaling Technology | 9271, RRID:AB_329825 | 1:1,000 Western blot |
| Antibody | Anti-Akt (rabbit polyclonal) | Cell Signaling Technology | 9272, RRID:AB_329827 | 1:1,000 Western blot |
| Antibody | Anti-p-S6 (rabbit polyclonal) | Cell Signaling Technology | 2211, RRID:AB_331679 | 1:1,000 Western blot |
| Antibody | Anti-S6 (rabbit polyclonal) | Cell Signaling Technology | 2217, RRID:AB_331355 | 1:1,000 Western blot |
| Antibody | Anti-Actin (mouse monoclonal) | Sigma-Aldrich | A2228, RRID:AB_476697 | 1:10,000 Western blot |
| Sequence-based reagent | quantitative PCR primers | Integrated DNA technologies | N/A | Custom DNA oligos |
| Peptide, recombinant protein | Collagenase D | Roche | COLLD-RO | |
| Peptide, recombinant protein | Pronase | Roche | PRON-RO | |
| Peptide, recombinant protein | IL-1b | Peprotech | 211-11B | 10ng/mL |
| Commercial assay or kit | Diaminobenzidine (DAB) peroxidase kit | Vector Biolabs | SK4100 | |
| Commercial assay or kit | Lactate assay kit | Eton Biosciences | 1.2E+09 | |
| Commercial assay or kit | Purelink RNA Mini Kit | Ambion | 12183025 | |
| Commercial assay or kit | High capacity cDNA Reverse Transcription Kit | Applied Biosystems | 4368814 | |
| Commercial assay or kit | ATP assay kit | Biovision | K255 | |
| Commercial assay or kit | Luminescence assay kit | GoldBio | I-930 | |
| Commercial assay or kit | Glutamate-Glo Assay kit | Promega | J7021 | |
| Chemical compound and drugs | CB-839 | Selleck | S7655 | |
| Chemical compound and drugs | Rapamycin | MedChem Express | HY-10219 | |
| Chemical compound and drugs | Ammonium Chloride | Sigma-Aldrich | A9434 | |
| Chemical compound and drugs | L-glutamic acid | Sigma-Aldrich | G1626 | |
| Chemical compound and drugs | Streptavidin Horseradish peroxidase (HRP) | Vector Biolabs | SA-5004–1 | |
| Chemical compound and drugs | DAPI | Cell Signaling Technology | 9071 | |
| Chemical compound and drugs | Trizol | ThermoFisher | 15596026 | |
| Chemical compound and drugs | DCFDA | Sigma-Aldrich | D6883 | |
| Chemical compound and drugs | MTT | Sigma-Aldrich | M655 | |
| Chemical compound and drugs | Immunocal | Fisher Scientific | NC9044643 | |
| Chemical compound and drugs | iTaq universal SYBR Green | BioRad | 1725120 | |
| Software and algorithm | Gen5 software | Agilent BioTek | BTGENSCPRIM | |
| Software and algorithm | Prism | Graphpad | RRID:SCR_002798 | |

| Reagent type (species) or resource | Designation | Source or reference | Identifiers | Additional information |
|---|---|---|---|---|
| Software and algorithm | Wave | Agilent BioTek | RRID:SCR_014526 | |

## Animals

Male and female mice on C57BL/6 background were used. All the animals were housed at the Washington University School of Medicine barrier facility. All experimental protocols were carried out in accordance with the ethical guidelines approved by the Washington University School of Medicine Institutional Animal Care and Use Committee (approved protocol #21–0413).

## Murine cell culture

For murine chondrocyte experiments, chondrocytes were isolated from sterna of newborn pups (C57BL/6 J) age P1-P3 without consideration for sex. Cells were isolated by sequential digestion with pronase (2 mg/mL, PRON-RO, Roche) at 37°, followed by collagenase D (3 mg/mL, COLLD-RO, Roche) two times at 37°, and cultured in DMEM (Life Technologies, Carlsbad, CA, USA) containing 10% fetal bovine serum (FBS) and 1% penicillin/streptomycin (15140122, ThermoFisher, Waltham, MA, USA) and plated in tissue culture plates. For glutamine deprivation conditions, media was changed to high glucose DMEM containing glutamine or devoid of glutamine (Life Technologies, Carlsbad, CA, USA). For experiments, cells are treated with recombinant mouse IL-1β (211-11B, Peprotech, Cranbury, NJ, USA) at 10 ng/mL, CB-839 (S7655, Selleck, Chemicals, Houston, TX, USA), rapamycin (HY-10219, MedChem Express, Monmouth Junction, NJ, USA), ammonium chloride (A9434, Sigma-Aldrich, USA), L-asparagine (A0884, Sigma-Aldrich, St. Louis, MO, USA), or L-glutamatic acid (G1626, Sigma-Aldrich, St. Louis, MO, USA).

## Human cell culture

Cartilage fragments from discarded tissue post-surgery were collected in Dulbecco's Modified Eagle Medium: Nutrient Mixture F-12 (DMEM/F-12, Gibco, ThermoFisher, Waltham, MA, USA) containing 10% heat-inactivated FBS (Gibco, ThermoFisher, Waltham, MA, USA), 2% penicillin, and streptomycin (10,000 U/mL, Gibco, ThermoFisher, Waltham, MA, USA). Tissue fragments were digested using an enzyme cocktail containing 0.025% collagenase P (Roche, 1.5 U/mg) and 0.025% pronase (Roche, 7 U/mg) in complete DMEM/F-12 medium in a spinner flask. After incubation at 37°C for overnight, the digest was filtered through 70 μm pore-size cell strainer and centrifuged at 1500 rpm for 5 min. Pellet was washed with calcium- and magnesium-free Hank's Balanced Salt Solution (Gibco, ThermoFisher, Waltham, MA, USA) and suspended in complete DMEM/F-12 supplemented with 50 mg/L L-ascorbic acid.

## MLI model

MLI surgery was utilized to induce OA in mice. In this procedure, medial meniscus was surgically removed from the joint without disrupting patella or other ligaments. Sham surgery was performed on the contralateral joint in which a similar incision was made on the medial side without removal of the meniscus. After 2 weeks (acute phase), mice are sacrificed, and joints were collected for histology.

## Immunohistochemistry

Mouse long bones were harvested keeping knee joints intact and fixing in 10% neutral buffered formalin for 24 hr at room temperature followed by decalcification in Immunocal (StatLab, McKinney, TX) for 3 days with fresh Immunocal changed every 24 hr. Tissues were processed, embedded into paraffin, and sectioned 5 μm thick then stained with hematoxylin-eosin or safranin-O to visualize cartilage and bone. For immunohistochemistry, sections were deparaffinized and rehydrated using three changes of xylenes followed by ethanol gradient. Antigen retrieval in murine sections was performed by incubating samples in citrate buffer (pH 6.0) at 55°C overnight, followed by washing in PBS and subsequent quenching of endogenous peroxidase activity by incubation in 3% $H_2O_2$ for 15 min at room temperature. Sections were then blocked using blocking solution (10% normal goat serum, 5% BSA, and 0.1% Tween-20) for 1 hr at room temperature. Sections were incubated overnight at 4° with anti-ATF4 (10835–1-AP, ThermoFisher, Waltham, MA, USA, RRID:AB_2058600) antibody at a 1:200 dilution. Sections were rinsed in PBS several times followed by addition of 1:500 dilution

of biotinylated secondary antibody (BP-1100, Vector Biolabs, Burlingham, CA, USA) for 1 hr. Post-secondary antibody incubation, sections were washed with PBS-Tween or Phosphate buffered saline-Tween (PBS-T) several times followed by incubation with streptavidin-HRP (2 µg/mL) for 20 min. After extensive washing with PBS, sections were developed using DAB peroxidase kit (SK4100, Vector Biolabs, Burlingham, CA, USA), with development on each slide standardized to the same amount of time.

## Protein analysis by immunoblotting

Cell lysates for protein analysis were prepared by scraping cells in 1× Cell Lysis Buffer (Cell Signaling Technology, Danvers, MA, USA) containing 1× protease/phosphatase inhibitor (Thermo Fisher Scientific, Waltham, MA, USA; Halt Protease Phosphatase Inhibitor Cocktail). Blotting was performed using primary antibodies for LC3B (2775, CST, Danvers, MA, USA, RRID:AB_915950), p62 (2C11, Abnova, Taiwan, RRID:AB_437085), IκB-ζ (Cat# 14-16801-82, Invitrogen, RRID:AB_11218083), p-AKT (9271, CST, Danvers, MA, USA RRID: AB_329825), total Akt (9272, CST, Danvers, MA, USA, AB_329827), phospho-S6 (2211, CST, Danvers, MA, USA, RRID:AB_331679), total S6 (2217, CST, Danvers, MA, USA, RRID:AB_331355), and actin (Cat# A2228, Sigma, St. Louis, MO RRID:AB_476697). Protein concentration was determined by bicinchoninic acid assay (BCA) assay (23225, Pierce, ThermoFisher, Waltham, MA, USA), and equal amounts of protein were loaded onto SDS-PAGE gel. Representative images are displayed. Images were quantified using ImageJ software (RRID:SCR_003070). Images were inverted, and band pixel density was measured using 'measure' tool. Bands were normalized to housekeeping genes.

## Immunocytochemistry

Chondrocytes were plated on sterile, gelatin-coated glass coverslips placed in 24-well plates at lower concentration. Cells were cultured under normal media conditions, and treatments were performed in the 24-well plate. For staining, media was removed, and cells were fixed in 4% formaldehyde in PBS for 30 min. Cells were washed with PBS containing 0.1% saponin. Cells were blocked using blocking buffer (1× PBS, 5% normal goat serum (NGS), and 0.1% saponin) for 1 hr at room temperature. Cells were incubated with anti-LC3b (12741, CST, Danvers, MA, USA; RRID:AB_2617131) or anti-p62 (2C11, Abnova, Taiwan, RRID:AB_437085) antibodies at 1:100 concentration in antibody dilution buffer (1× PBS, 1% BSA, and 0.1% saponin) overnight at 4°. Cells were washed with wash buffer (1× PBS and 0.1% saponin) three times and incubated with fluorescent conjugated secondary antibody at 1:1000 in antibody dilution buffer for 2 hr at room temperature. Samples were washed with wash buffer three times. Slides were coverslipped with antifade mounting media containing DAPI (9071, CST, Danvers, MA, USA). Images were taken on fluorescent microscope.

## Measurement of extracellular lactic acid

Chondrocytes were cultured for 1 day with IL-1β treatment (10 ng/mL) with appropriate experimental conditions in 96-well plates containing 200 µL of DMEM containing 10% FBS. Supernatant media was collected and centrifuged to separate cell debris and floating cells. Supernatant was used immediately for lactic acid assay to measure secreted lactate in the media using a 1:20 dilution (Cat# 1200011002, Eton Biosciences, San Diego, CA, USA). Unconditioned DMEM with 10% FBS was used as a control for subtracting background.

## Measurement of gene expression by qPCR

Trizol (Sigma, St. Louis, MO, USA) was added to samples to isolate mRNA from cell culture samples. Chloroform was added at a ratio of 0.2:1 to Trizol to samples, followed by centrifugation at 12,000 g for 15 min. Aqueous layer was isolated, and equal amount of 70% ethanol was added. RNA was then isolated from this fraction using PureLink RNA mini kit (Cat# 12183025, Ambion, Grand Island, NY, USA), and cDNA was prepared using High Capacity cDNA Reverse Transcription kit (Cat# 4368814, Applied Biosystems). Quantitative PCR (qPCR) was carried out on BioRad CFX96 real time system using iTaq universal SYBR green super-mix (Cat#1725120, BioRad, Hercules, CA, USA). mRNA expression was normalized using actin as a housekeeping gene. Full list of primers is listed in *Table 1*.

## Measurement of cellular metabolism by Seahorse

Primary chondrocytes were plated in Seahorse XF96 plates at 50,000 cells per well and cultured in media containing glutamine or without glutamine. Cells were then treated with IL-1β (10 ng/mL). After

**Table 1.** List of primers.

| Primer | Sequence (5′→3′) |
| --- | --- |
| m-Il6 | GCTACCAAACTGGATATAATCAGGA |
| | CCAGGTAGCTATGGTACTCCAGAA |
| m-Mmp13 | GCCAGAACTTCCCAACCAT |
| | TCAGAGCCCAGAATTTTCTCC |
| m-Atf4 | TCGATGCTCTGTTTCGAATG |
| | AGAATGTAAAGGGGGCAACC |
| m-Lc3 | TGGGACCAGAAACTTGGTCT |
| | GACCAGCACCCCAGTAAGAT |
| m-p62 | AGAATGTGGGGGAGAGTGTG |
| | TCTGGGGTAGTGGGTGTCAG |
| m-GLS | CTACAGGATTGCGAACATCTGAT |
| | ACACCATCTGACGTTGTCTGA |
| m-GDH | GGCCGATTGACCTTCAAATA |
| | TCCTGTCCTGGAACTCTGCT |
| m-GS | CATTGACAAACTGAGCAAGAGG |
| | AAGTCGTTGATGTTGGAGGTT |
| m-EAAT2 | GGCAATCCCAAACTCAAGAA |
| | GTGCTATTGGCCTCCTCAGA |
| m-ASCT2 | CAACCAAAGAGGTGCTGGAT |
| | CCTCCACCTCACAGAGAAGC |
| m-G6pd2 | CTGAATGAACGCAAAGCTGA |
| | CAATCTTGTGCAGCAGTGGT |
| m-Eno1 | GCCTCCTGCTCAAAGTCAAC |
| | AACGATGAGACACCATGACG |
| m-Ldha | TGGAAGACAAACTCAAGGGCGAGA |
| | TGACCAGCTTGGAGTTCGCAGTTA |
| m-Mdh | GGTGCAGCCTTAGATAAATACGC |
| | AGTCAAGCAACTGAAGTTCTCC |
| m-Sdha | AACACTGGAGGAAGCACACC |
| | AGTAGGAGCGGATAGCAGGA |
| m-Idh2 | AACCGTGACCAGACTGATGAC |
| | ATGGTGGCACACTTGACAGC |
| m-Got2 | GATCCGTCCCCTGTATTCCA |
| | CACCTCTTGCAACCATTGCTT |
| h-GLS2 | TCTCTTCCGAAAGTGTGTGAGC |
| | CCGTGAACTCCTCAAAATCAGG |
| h-GLUD1 | TATCCGGTACAGCACTGACG |
| | GCTCCATGGTGAATCTTCGT |
| h-GS | CCTGCTTGTATGCTGGAGTC |

*Table 1 continued on next page*

*Table 1 continued*

| Primer | Sequence (5′→3′) |
| --- | --- |
|  | GATCTCCCATGCTGATTCCT |
| h-GOT2 | GTTTGCCTCTGCCAATCATATG |
|  | GAGGGTTGGAATACATGGGAC |
| h-NFKBIZ | CCGTTTCCCTGAACACAGTT |
|  | AGAAAAGACCTGCCCTCCAT |
| h-MMP3 | CTGGACTCCGACACTCTGGA |
|  | CAGGAAAGGTTCTGAACTGACC |
| h-ATF4 | TCTCCAGCGACAAGGCTAA |
|  | CAATCTGTCCCGGAGAAGG |

24 hr, Seahorse assay was performed. For glycolysis stress test, cells were serum starved for 1 hr in glucose-free media containing treatments, and measurement of ECAR and OCR was performed prior to and after sequential addition of glucose, oligomycin, and 2-DG with measurements performed every 5 min. For MitoStress test, cells were incubated in glucose-containing media for 1 hr containing treatments, and measurements were performed every 5 min prior to and after sequential addition of oligomycin, FCCP, and rotenone/antimycin A. Media for Seahorse assays was devoid of glutamine. Data was analyzed using Wave software.

## Measurement of intracellular ATP

Primary chondrocytes were plated in 96-well plates at $5 \times 10^4$ and treated with IL-1β for 24 hr. Lysates were collected and processed according to luminescence-based ATP assay kit (Cat#K255, Biovision, Milpitas, CA, USA; ADP/ATP ratio Assay kit). Assay was performed in 96-well plate. Luminescence was measured using luminescent plate reader after 15 min. Data was collected and processed using Gen5 software.

## Measurement of ROS

Primary chondrocytes were treated for 24 hr in DMEM media. Cells were washed two times with phenol red-free PBS, followed by incubation with 10 µM DCFDA (Cat#D6883, Sigma, St. Louis, MO, USA) in PBS for 30 min, followed by two more washes with PBS. Cells were incubated in 37°C incubator for 1 hr in PBS, followed by fluorescence measurement using microplate reader using Ex/Em 495/525 for DCFDA.

## Measurement of metabolite concentrations

The cell suspensions ($2 \times 10^6$ cells/mL) were prepared by vortexing cell pellets with water. The amino acids and metabolites listed above were extracted from 50 µL of cell suspension with 200 µL of methanol after addition of internal standards (Glu-d3 [1.6 µg], Asp-d3 [1.6 µg], Asn-d3,15N2 [1.6 µg], Gln-13C5 [1.6 µg], alpha-ketoglutarate-d2 [0.4 µg], 2-hydroxyglutarate-d3 [0.2 µg], oxaloacetate-13C3 [0.2 µg], pyruvate-13C3 [2 µg], and malate-d3 [0.2 µg]). The sample aliquots for alpha-ketoglutarate, oxaloacetate, and pyruvate were derivatized with o-phenylenediamine to improve mass spectrometric sensitivity. Quality control (QC) samples were prepared by pooling aliquots of study samples to monitor instrument performances throughout these analyses.

The analysis of Gln, Glu, Asp, and Asn was performed on a Shimadzu 20AD HPLC system and a SIL-20AC autosampler coupled to 4000Qtrap mass spectrometer (Applied Biosystems) operated in positive multiple reaction monitoring (MRM) mode. The analysis of 2-hydroxyglutarate and malate was performed on a Shimadzu 20AD HPLC system and a SIL-20AC autosampler coupled to 4000Qtrap mass spectrometer (Applied Biosystems) operated in negative MRM mode. The analysis of alpha-ketoglutarate, oxaloacetate, and pyruvate was performed in positive ion MRM mode on API4000 mass spectrometer (Applied Biosystems) coupled to a Shimadzu 20AD HPLC system and a SIL-20AC

autosampler. The QC samples were injected every five study samples. Data processing was conducted with Analyst 1.6.3 (Applied Biosystems).

## Measurement of viability

Cells were treated under appropriate conditions. MTT (3-[4,5-dimethylthiazol-2-yl]–2,5-diphenyltetrazolium bromide) was dissolved in PBS at 10 mg/mL. MTT was added to wells to a final concentration of 0.5 mg/mL. Cells were incubated at 37° for 6 hr. Media was removed, and 50 uL of DMSO was added to each well. Plate was placed on a plate shaker, and measurement was made at 580 nm on microplate reader. Data was collected and processed using Gen5 software.

## Measurement of intracellular glutamate

Intracellular glutamate was measured in chondrocytes in 96-well plate format utilizing Glutamate-Glo assay kit (J7021, Promega, Madison, WI, USA). Luminescence was measured on microplate reader. Data was collected and processed using Gen5 software.

## NF-κB luciferase assay

Chondrocytes were isolated from sterna of newborn NF-κB-luciferase reporter mice as described earlier. Cells were cultured and treated appropriately in 96-well plate tissue culture plate. Cells were washed twice with 1× PBS and lysed using 1× luciferase lysis buffer (L-740, GoldBio, St. Louis, MO, USA). Plates were freeze-thawed in –80 freezer. 20 μL of lysates were transferred to white bottom, round well microplates. Detection was performed after addition of 50 μL of detection reagent (I-930, GoldBio, St. Louis, MO, USA). Luminescence was measured by microplate reader. Data was collected and processed using Gen5 software.

## Statistical analysis

All experiments represent biological replicates and were repeated at least three times, unless otherwise stated. Technical replicates are considered to be repeat tests of the same value, i.e., testing same samples in triplicate for qPCR. Biological replicates are considered to be samples derived from separate sources, such as different mice or on different dates. Statistical analyses were performed using appropriate statistical test using GraphPad Prism. All graphs were generated using Prism as well. Multiple treatments were analyzed by one-way ANOVA followed by Tukey's test multiple comparisons test for greater than two groups. Student's T test was used for comparing two groups. Student's T test was performed for comparing same biological samples subject to different treatments. p-Values are indicated where applicable. $*p<0.05$, $**p<0.01$, $***p<0.0005$, and $****p<.0001$. Histology and immunostaining data were scored by investigators blinded to the experimental conditions. Male and female mice were used at equal ratios for cell culture to avoid sex bias. Sample size determination was not required.

## Acknowledgements

This work is supported by NIH/NIAMS R01-AR072623 (to YA), Biomedical grant from Shriners Hospital for Children (YA), P30 AR074992 NIH Core Center for Musculoskeletal Biology and Medicine (to YA), R01-AR076758 and R01-AI161022 (to GM).

## Additional information

### Funding

| Funder | Grant reference number | Author |
| --- | --- | --- |
| National Institute of Arthritis and Musculoskeletal and Skin Diseases | AR072623 | Yousef Abu-Amer |

| Funder | Grant reference number | Author |
| --- | --- | --- |
| National Institute of Arthritis and Musculoskeletal and Skin Diseases | AR074992 | Yousef Abu-Amer |
| Shriners Hospitals for Children | 85160-STL-20 | Yousef Abu-Amer |
| National Institute of Arthritis and Musculoskeletal and Skin Diseases | AR076758 | Gabriel Mbalaviele |
| National Institutes of Health | AI161022 | Gabriel Mbalaviele |

The funders had no role in study design, data collection and interpretation, or the decision to submit the work for publication.

### Author contributions

Manoj Arra, Data curation, Conceptualization, Formal analysis, Investigation, Writing - original draft, Writing - review and editing; Gaurav Swarnkar, Conceptualization, Formal analysis, Investigation; Naga Suresh Adapala, Syeda Kanwal Naqvi, Conceptualization; Lei Cai, Conceptualization, Methodology; Muhammad Farooq Rai, Resources, Conceptualization; Srikanth Singamaneni, Formal analysis, Methodology, Writing - review and editing; Gabriel Mbalaviele, Resources, Formal analysis, Funding acquisition; Robert Brophy, Resources; Yousef Abu-Amer, Data curation, Resources, Formal analysis, Supervision, Funding acquisition, Validation, Writing - original draft, Project administration, Writing - review and editing

### Author ORCIDs

Muhammad Farooq Rai http://orcid.org/0000-0003-4826-4331
Gabriel Mbalaviele http://orcid.org/0000-0003-4660-0952
Yousef Abu-Amer http://orcid.org/0000-0002-5890-5086

### Ethics

This study was performed in strict accordance with the recommendations in the Guide for the Care and Use of Laboratory Animals of the National Institutes of Health. All of the animals were handled according to approved institutional animal care and use committee (IACUC) protocols (#21-0413) of Washington University. All surgery was performed under sodium pentobarbital anesthesia, and every effort was made to minimize suffering.

### Decision letter and Author response

Decision letter https://doi.org/10.7554/eLife.80725.sa1
Author response https://doi.org/10.7554/eLife.80725.sa2

## Additional files

### Supplementary files

• MDAR checklist

### Data availability

All data generated or analysed during this study are included in the manuscript and supporting file. Source data have been provided for all figures.

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
