## [Editor Report]

This manuscript focuses on identifying how metabolism can influence the response of cartilage cells to inflammation. This has a relevance to the painful disease known as osteoarthritis. Modulation of call metabolism in the right direction can serve to protect joint cartilage from the negative effects of inflammation with causes onset and disease progression.

---

## [Decision Letter]

**Decision letter after peer review:**

Thank you for submitting your article "Glutamine metabolism modulates chondrocyte inflammatory response" for consideration by *eLife*. Your article has been reviewed by 3 peer reviewers, one of whom is a member of our Board of Reviewing Editors, and the evaluation has been overseen by Mone Zaidi as the Senior Editor. The reviewers have opted to remain anonymous.

Essential revisions:

Several of the reviewers have noted items that hinder data interpretation. Please conduct a revision to correct these items to enhance the overall manuscript.

*Reviewer #1 (Recommendations for the authors):*

Overall the manuscript is excellent, thus there are no significant comments. Stylistically, several times the authors note that they were surprised by certain findings. As the readership may not have the underlying context for those surprises, perhaps adding explanatory sentences with references will better help "set the stage" so to speak for those sections.

*Reviewer #3 (Recommendations for the authors):*

A few points need to be taken into consideration in order to make this manuscript suitable for publication.

Figure 3F shows that ammonia stabilizes IkBz protein expression, not IkBz (Nfkbiz) gene expression as stated on page 8 line 11. This should be corrected. Related to this, the conclusion on page 8 lines 16 and 17 should be extended to include: "promoting inflammation through stabilization of IkBz and its transcriptional inflammatory response".

On page 8 line 18, the paragraph describes a modest inhibition of the inflammatory response by EGCG (reduced a-KG) compared to glutamine deprivation. This should be further discussed to define potential differences between these inflammatory responses.

The authors show that glutamine deprivation had opposing effects on mTOR1 and mTOR2 activation. Rapamycin, which is primarily a mTOR1 inhibitor, was shown to be pro-inflammatory. The authors should further speculate on the impact of mTOR2 inhibition on the inflammatory response in the context of glutamine deprivation.

Can the effect of glutamine deprivation be rescued with asparagine supplementation, since both can liberate ammonia?

---

## [Author Response]

Essential revisions:Reviewer #1 (Recommendations for the authors):Overall the manuscript is excellent, thus, there are no significant comments. Stylistically, several times the authors note that they were surprised by certain findings. As the readership may not have the underlying context for those surprises, perhaps adding explanatory sentences with references will better help "set the stage" so to speak for those sections.

We addressed the conflict with recent publications in the discussion and introduction of the manuscript. We believe that there are several possible explanations for the differing results. Importantly, our work focuses on glutamine deprivation while the paper by Ma et al., 2022 utilizes glutamine supplementation to supra-physiological levels. It’s possible that both of these approaches can activate protective pathways that may be useful for the treatment of OA. We also added explanatory background sentences to better explain why we may have been “surprised” at some results. For example, we explain why it was unexpected that glutamine deprivation would reduce oxidative stress on line 8 of page 8. Overall, there is further work that needs to be done to explore these unexpected findings in further mechanistic details.

Reviewer #3 (Recommendations for the authors):A few points need to be taken into consideration in order to make this manuscript suitable for publication.Figure 3F shows that ammonia stabilizes IkBz protein expression, not IkBz (Nfkbiz) gene expression as stated on page 8 line 11. This should be corrected. Related to this, the conclusion on page 8 lines 16 and 17 should be extended to include: "promoting inflammation through stabilization of IkBz and its transcriptional inflammatory response".On page 8 line 18, the paragraph describes a modest inhibition of the inflammatory response by EGCG (reduced a-KG) compared to glutamine deprivation. This should be further discussed to define potential differences between these inflammatory responses.The authors show that glutamine deprivation had opposing effects on mTOR1 and mTOR2 activation. Rapamycin, which is primarily a mTOR1 inhibitor, was shown to be pro-inflammatory. The authors should further speculate on the impact of mTOR2 inhibition on the inflammatory response in the context of glutamine deprivation.Can the effect of glutamine deprivation be rescued with asparagine supplementation, since both can liberate ammonia?

The writing regarding the results of figure 3F were re-written to appropriately refer to gene expression.

The difference between EGCG and glutamine deprivation was also further expounded upon. It is clear that EGCG targets an enzyme multiple steps downstream of glutamine breakdown, hence it is unlikely to have the same broad signaling effect that glutamine deprivation will have. Furthermore, glutamine can contribute to many different substrates, while EGCG primarily blocks aKG production.

We addressed to some degree the different functions of mTOR1 and mTOR2 and speculated about why rapamycin may be pro-inflammatory. Unfortunately, mTOR2 is understudied, especially in the context of inflammation. This will be the focus of future work, as we interrogate the different functions of mTOR1 and mTOR2 in chondrocytes. However, we included several references that describe the functions of mTOR1 and mTOR2 in the context of metabolism and immunity.

We included a figure showing that asparagine supplementation was unable to rescue glutamine deprivation, and explain why we believed it may not be able to. It is potentially related to some unique function of amino acid sensing machinery and interchangeability of ammonia and substrates from asparagine vs glutamine.

Language was modified to be more scientific.